# Anemia prevalence and severity among pregnant refugee women settled in the West Nile region, Uganda

Felix Bongomin[1]*, Winnie Kibone[1], Ritah Nantale[2], Sarah Lebu[3],
Byron Awekonimungu[1], Phillip Musoke[1], Daniel S. Ebbs[4], Rachel Beardsley[3],
Chimdi Muoghalu[3], Stephen Ochaya[1], Musa Manga[3]

**1** Department of Medical Microbiology and Immunology, Faculty of Medicine, Gulu University, Gulu, Uganda, **2** Faculty of Health Sciences, Busitema University, Mbale, Uganda, **3** Department of Environmental Sciences and Engineering, Gillings School of Public Health, University of North Carolina at Chapel Hill, North Carolina, United States of America, **4** Section of Critical Care Medicine, Department of Paediatrics, Yale University, New Haven, Connecticut, United States of America

\* drbongomin@gmail.com

## Abstract

### Background

Anemia during pregnancy is a significant global public health issue, associated with adverse outcomes for both the mother and the newborn. While the prevalence and impact of anemia in pregnancy have been well-documented in general populations, the burden and severity of anemia among pregnant refugee women particularly those residing in the West Nile region of Uganda, remain unknown. Our study aimed to determine the prevalence and severity of anemia in this specific population to address this critical knowledge gap.

### Methods

Between 1st April and 30th June 2023, we conducted a cross-sectional study among pregnant women attending the antenatal care clinic at 3 health centers serving the refugee communities in Adjumani district, Uganda. Anemia in pregnancy was defined as a hemoglobin (Hb) level of < 11.0 g/dl and microcytosis as a mean corpuscular volume of < 76 fL. Multivariable logistic regression was done to assess for the independent predictors of anemia in pregnancy. Data were analyzed in Stata version 15.0.

### Results

We enrolled 304 participants. The mean age of the participants was 25.4 years with a standard deviation of 4.9 years. The prevalence of anemia was 52.6% (95% CI 46.9–58.4). Of these, 85 (28.0%) were classified as mild, 73 (24.1%) moderate, and 2 (0.7%) had severe anemia. The median Hb was 10.8g/dL (10.1–11.7). Forty-one

**Data availability statement:** All relevant data are within the paper and its Supporting Information files.

**Funding:** The author(s) received no specific funding for this work.

**Competing interests:** The authors have declared that no competing interests exist.

(25.6%) of those with anemia had microcytosis. Having at least a secondary/tertiary education level (aOR: 1.51, 95%CI: 0.98–1.78, p < 0.001), being single (aOR: 1.77, 95%CI: 1.53–1.93, p < 0.001) and parity >4 (aOR: 1.51, 95%CI: 0.98–1.78, p < 0.001) were associated with higher odds of having anemia.

## Conclusions

Anemia in pregnancy is prevalent among pregnant refugee women in the West Nile Region of Uganda. Our findings suggest the need for targeted interventions such as health education about the consequences of anemia, promotion of early antenatal care (ANC) attendance, and provision of iron and folate supplementation during pregnancy, to potentially reduce the burden of anemia in this vulnerable population.

## Introduction

Anemia in pregnancy, defined as hemoglobin concentration levels of less than 11 g/dl is a major public health problem [1,2]. In 2019, the global prevalence of anemia among pregnant women was 36.5% [3]. Low- and middle-income countries (LMICs) bear a higher burden of anemia in pregnancy with prevalence estimates ranging from 24% to 56%, compared to 22% (16–29%) in high-income countries [2]. In the East African region, similar to other LMICs, the prevalence of anemia in pregnancy remains high, estimated at 36% (30–41%) [2].

Anemia in pregnancy is associated with poor maternal and perinatal outcomes [4,5]. In LMICs, it contributes to 17% of low birthweight, preterm birth and perinatal mortality cases [2]. The increased iron demands in pregnancy due to the expansion of red cell mass, placenta and fetal growth, predispose pregnant women to anemia [6–8]. In humanitarian settings, such as refugee camps, the burden of anemia in pregnancy can be exacerbated by factors like limited access to healthcare, poor nutrition, and high rates of infectious diseases [9]. Pregnant refugee women might be disproportionately affected due to these compounded vulnerabilities, which can lead to higher rates of anemia compared to the general population.

The Ministry of Health in Uganda recommends iron and folate supplementation, malaria prevention and control with intermittent preventive treatment (IPT) and use of insecticide-treated mosquito nets, prevention and treatment of hookworm infestation with dewormers and routine screening for anemia during antenatal care to control and treat anemia during pregnancy [10]. However, compliance with these measures remains poor, for instance a study by Kiwanuka *et al* at Mulago national referral hospital showed that only 12% of women attending antenatal care were compliant with iron and folate supplementation [11].

Previous studies on anemia in pregnant refugee women have reported varying prevalence rates and identified factors such as poor dietary intake, limited healthcare access, and high infection rates as drivers of anemia. For example, prevalence rates have been reported as 36.1% among South Sudanese refugees in Ethiopia [12], 38.6% among Palestinian refugees [13], 36.3% among South Sudanese refugees in

Uganda [14], and 35.0% in Malaysia [15]. A recent systematic review of over 14,000 Ugandan pregnant women showed a prevalence of anemia of about 30% [16]. Despite this high burden, few studies have established the prevalence, associated factors, and consequences of anemia in pregnancy specifically in Uganda. Whether the prevalence of anemia among refugee pregnant women settled in Uganda differs from the general population remains unclear. The aim of this study, therefore, was to determine the prevalence and severity of anemia among pregnant refugee women settled in the West Nile region of Uganda. This study sought to address the existing gap in the literature regarding the burden of anemia in this vulnerable population and to identify specific factors associated with anemia in this context. By doing so, we aim to inform targeted interventions that can help reduce the burden of anemia among pregnant refugee women.

## Methods

### Study design

Between 1st April April and 30th June 2023, we conducted a multi-center, descriptive cross-sectional study to determine the prevalence and severity of anemia among pregnant refugee women in adjumani district, Uganda.

### Study settings

Adjumani district has 41 health facilities, including 12 within the settlements. We randomly selected three health facilities serving refugee communities in Adjumani district, namely, Mungula Health Centre (HC) IV, Ayiri HC III, and Pagirinya HC III. These facilities serve both the refugee and the host population. The ANC clinics in all the study sites run from Monday to Friday every week, offering ANC services to about 20–40 mothers every clinic day. Services offered at the ANC includes, health education, malaria screening, HIV testing, intermittent malaria prophylaxis, iron and folate supplementation, deworming, and immunization.

### Participants and sample size

We enrolled pregnant women aged 18 years or older who were willing and competent to provide informed written consent, regardless of gestation age or gravidity. Women living with sickle cell anemia and host population were excluded. With an estimated prevalence of anemia among pregnant women in Uganda of 30% [16], precision of 5%, a two-tailed Z-statistic of 1.96, a sample size of 322 participants was calculated using the Kish-Leslie formula [17]. However, we able to enroll 304 participants due to logistic challenges within the settlement. Trained research assistants consecutively enrolled eligible participants until the sample size was realized. Questionnaires were administered in several languages including the local language (*Ma'di*), English, and Arabic with the help of trained translators. Data collection was supervised by two investigators (FB and WK) and data entry was continuously monitored by SL for completeness using Kobo Toolbox.

### Study measurements

**Participants characteristics.** Data were collected through surveys administered by trained research assistants on information on maternal characteristics such as age, gravidity, education level, occupation, marital status, HIV status, and the number of ANC visits in the current pregnancy. Study variables to include in the questionnaire were guided by previous studies in Uganda. HIV statuses were confirmed from the ANC card. Data on income, deworming and other socio-behavioral characteristics were self-reported by the participants.

**Blood collection.** A single 4 mL venous blood sample was collected from each participant by trained study nurses using EDTA vacutainers (lavender top) (BD, Franklin Lakes, NJ) through standard aseptic venipuncture techniques. This volume was selected to meet the minimum requirements for both complete blood count (CBC) analysis and malaria testing while minimizing participant burden. Immediately after collection, samples were gently inverted 8–10 times to ensure proper anticoagulant mixing, labeled with unique study identifiers, and placed in temperature-controlled cool boxes

(4°C) for transport. Samples were processed at Mungula HCIV laboratory within 4 hours of collection, where the blood was aliquoted for parallel CBC analysis and malaria diagnostic testing.

**Complete blood count.** All specimens were processed using the HumaCount 3D Hematology System (HUMAN GmbH, Wiesbaden, Germany), with analysis completed within the 4-hour stability window for complete blood count testing. Standard operating procedures followed Ugandan Ministry of Health guidelines, including: [1] chain-of-custody documentation at each transfer point, [2] continuous temperature monitoring during transport, and [3] visual inspection for hemolysis or clots prior to analysis. The laboratory implemented daily quality controls per manufacturer specifications to ensure instrument accuracy. Anemia in pregnancy was defined according to the WHO classification as Hb < 11g/dl and further classified into mild (Hb 10.0–10.9 g/dL), moderate (Hb 7.0 – 9.9g/dL) and severe (Hb < 7.0g/dL. Mean corpuscular volume of <76 femtolitre was considered microcytic, 76–100 normocytic and >100 macrocytic [18].

**Anthropometry.** Weight was measured with minimal clothing and without shoes using a digital bathroom weighing scale (SECA-Germany).

**Malaria test.** All participants underwent simultaneous malaria testing using both microscopy and rapid diagnostic tests (RDTs) from a single blood sample. For microscopy, thick blood smears were prepared, stained with 3% Giemsa for 45 minutes, and independently examined by two blinded laboratory technologists under 1000 × oil immersion microscopy. Parasite density was calculated by counting against 200 WBCs (or 500 WBCs for low-density infections) using an 8000 WBCs/μL conversion factor. Parallel RDT testing employed CareStart Malaria HRP-2/pLDH (Pf/Pan) combo tests (Access Bio, Belgium), which detect P. falciparum via HRP-2 and other Plasmodium species through pan-pLDH. Stored at room temperature with verified expiration dates, tests were performed per manufacturer instructions, interpreted at 15 minutes, and repeated if control lines were absent. All results were tracked using consistent participant IDs across laboratory records and clinical data, with non-falciparum infections identified by isolated pan-pLDH band positivity.

## Statistical analysis

Data were downloaded from KoboToolbox and exported as a Microsoft Excel document for cleaning and coding. Categorical variables were expressed as frequencies and percentages. Parametric data were summarized as mean and standard deviations (mean±SD) and non-parametric data as median and interquartile range. Multivariable logistic regression model was used to assess for independent predictors of anemia in pregnancy. All variables with $p < 0.2$ at the bivariable level were fitted into a multi-variable logistic regression model and adjusted for potential confounders such as age, parity. Stata version 15.0 (StataCorp LLC) was used for data analysis. Multicollinearity among the independent variables was checked using the variance-inflation factor. Adjusted odds ratio (aOR) and 95% CI were used to estimate strength of association. Statistical significance was set at $p < 0.05$.

## Ethical considerations

The study protocol was approved by the Gulu University Research and Ethics Committee (GUREC-2022–450, 10th March 2023), and additional administrative approvals were obtained from the Office of the Prime Minister (Ref# RDA/1B/081/23), Medical Teams International (MTI) and at all the study sites. All study participants provided written informed consent. Principles of human subject research outlined in the *Declaration of Helsinki* were strictly adhered to. All participants were provided with a copy of their CBC, and malaria tests, and those with positive findings were provided appropriate care in the routine clinical settings. Further, the research team was trained in research ethics principles.

## Results

### Participant characteristics

**Sociodemographic characteristics of the participants.** A total of 304 participants were included in the final analysis, a response rate of 94.4%. The mean age of the participants was 25.4 years with a standard deviation of 4.9 years. Most

participants were in the age group of 25–34 years (48.4%, n = 147), were married (93.8%, n = 285) and had attained a primary level of education (61.8%, n = 188). Regarding source of income, majority (85.9%, n = 261) reported that they had no source of income and almost all participants (97.0%, n = 295) had an average monthly household income of less than UGX 100,000 (27 USD). Most (39.7%, n = 118) participants spent between 1–3 years at the camp. Details are in Table 1.

### Obstetric and medical characteristics of the participants

Most (35.5%, n = 108) participants had parity of 2–4. Seventeen (5.6%) reported history of an abortion. Regarding ANC attendance, the majority had attended ANC less than four times (64.5%, n = 196). Almost all participants (99.0%, n = 300) were HIV negative and 16 (5.3%) had a positive malaria rapid diagnostic test (mRDT) and 3.2% (n = 10; seven *P. falciparum* and three *P. malariae*) by microscopy, all were 1–10 parasites per 100 thick film fields (Table 2).

### Prevalence of anemia among pregnant women living in refugee camps in Northern Uganda

The prevalence of anemia was 52.6% (95% CI 46.9–58.4). Of these, 85 (28.0%) were classified as mild, 73 (24.1%) moderate, and 2 (0.7%) had severe anemia. The median Hb was 10.8g/dL (10.1–11.7). Forty-one (25.6%) of those with anemia had microcytosis (mean corpuscular volume [MCV] < 76 fL).

### Factors associated with anemia among pregnant women living in refugee camps in Northern Uganda

Having a secondary/tertiary education level (aOR: 1.13, 95%CI: 1.10–1.16, p < 0.001), being single (aOR: 0.51, 95%CI: 0.46–0.58, p < 0.001) and parity >4 (aOR: 1.49, 95%CI: 1.10–2.01, p < 0.001) were associated with higher odds of having anemia (Table 3).

## Discussion

In this study, we assessed the prevalence of anemia and associated factors among pregnant refugee women settled in adjumani district, West Nile Region, Uganda. We found a prevalence of anemia of 52.6%, with most 28.0% having mild anemia, 24.1% moderate, and 2 (0.7%) severe anemia.

The prevalence of anemia in our study was high as compared to the 30% reported by a systematic review conducted among pregnant women in Uganda [16]. However, majority of the studies included in this systematic review were done in central Uganda, an urban area with a higher socioeconomic status, in contrast to our study done in rural northern Uganda. Additionally, our study included refugee pregnant women who face various challenges including cultural and language barriers, discrimination, and financial hardships which may result in missed antenatal care visits [19]. Our finding was also higher than the anemia prevalence reported among pregnant refugee women in Ethiopia (36.1%), Rwanda (20.8%) and Somalia (44.4%) [20–22]. Nevertheless, our prevalence was comparable to the 53% and 59% reported among pregnant women in Sudan [23] and India [24] but lower than the 78% reported among pregnant women in Ghana [25] respectively. The variations in the reported prevalence rates across studies could be due to the differences in the cut-off values for haemoglobin levels of the study participants, study setting and population.

In our study, a quarter of the women with anemia had microcytosis. Similar findings have been observed in other studies done among pregnant women [16,26]. Microcytic anemia during pregnancy could be linked to the insufficient intake of iron supplements despite the increased iron demands associated with pregnancy [16]. Our findings highlight gaps in the implementation of anaemia prevention and control measures among pregnant refugee women.

We found out that pregnant women with a secondary/tertiary education level had 1.5 times the odds of having anemia in pregnancy compared with those with none/primary education level. Similarly, Novivanti *et al.*, 2019 revealed that less-educated pregnant women are less likely to be anaemic than highly educated ones [27]. Contrary to our finding, most studies have showed that less educated women have higher odds of being anaemic as compared to those who are highly educated [27–30]. Educated women may delay seeking antenatal care due to work commitments or perceive themselves

**Table 1. Sociodemographic characteristics of the participants.**

| Characteristic (n = 304) | Frequency (n) | Percentage (%) |
|---|---|---|
| **Study site** | | |
| Ayiri HC III | 54 | 17.8 |
| Mugula health centre IV | 101 | 33.2 |
| Pagirinya health clinic III | 149 | 49 |
| **Age (years)**, Mean±SD; 25.42 ± 4.85 years | | |
| 15–24 | 139 | 45.7 |
| 25–34 | 147 | 48.4 |
| ≥35 | 18 | 5.9 |
| **Highest education level** | | |
| Informal | 29 | 9.5 |
| Primary | 188 | 61.8 |
| Secondary | 67 | 22 |
| Tertiary | 20 | 6.6 |
| **Marital Status** | | |
| Married | 285 | 93.8 |
| Separated | 14 | 4.6 |
| Widowed | 5 | 1.6 |
| **Participant is the bread winner of the household** | | |
| No | 37 | 12.2 |
| Yes | 267 | 87.8 |
| Source of income | | |
| Business | 38 | 12.5 |
| Casual labourer | 1 | 0.3 |
| Formal employment | 4 | 1.3 |
| I have no source of income | 261 | 85.9 |
| **Estimated monthly income (USD)**, Median (IQR); 19 (15–19) | | |
| <27 | 295 | 97 |
| 27–135 | 8 | 2.6 |
| >135 | 1 | 0.3 |
| **Time at the camp** | | |
| Less than 1 year | 88 | 29.6 |
| 1 to 3 years | 118 | 39.7 |
| >3 years | 91 | 30.6 |
| **Share water source with animals** | | |
| No | 295 | 97 |
| Yes | 9 | 3 |
| **Engaged in farming** | | |
| No | 74 | 24.3 |
| Yes | 230 | 75.7 |
| **Engaged in pig farming** | | |
| No | 184 | 60.5 |
| Yes | 120 | 39.5 |
| **Share home with animals** | | |
| No | 101 | 33.2 |
| Yes | 203 | 66.8 |
| **Weight**, Mean±SD; 60.22 ± 5.49 kgs | | |

**Table 2. Obstetric and medical characteristics of the participants.**

| Characteristic (n = 304) | Frequency (n) | Percentage (%) |
|---|---|---|
| Gravidity, median (IQR) 2 (1–3) | | |
| Primigravida | 78 | 25.7 |
| 2–4 | 186 | 61.2 |
| >4 | 40 | 13.2 |
| Parity, median (IQR) 1(0–2) | | |
| 0 | 86 | 28.3 |
| 1 | 92 | 30.3 |
| 2–4 | 108 | 35.5 |
| >4 | 18 | 5.9 |
| History of Abortion | | |
| No | 287 | 94.4 |
| Yes | 17 | 5.6 |
| Number of ANC visits | | |
| <4 | 196 | 64.5 |
| ≥4 | 108 | 35.5 |
| Has Gastritis/ Ulcers | | |
| No | 262 | 86.2 |
| Yes | 42 | 13.8 |
| HIV status | | |
| Negative | 300 | 99 |
| Positive | 3 | 1 |
| HPV infection or cervical cancer | | |
| No | 298 | 98.3 |
| Yes | 5 | 1.7 |
| Diabetes mellitus history | | |
| No | 299 | 98.7 |
| Yes | 4 | 1.3 |
| Dewormed in the last 6 months | | |
| No | 274 | 90.1 |
| Yes | 30 | 9.9 |
| Had positive mRDT | | |
| No | 288 | 94.7 |
| Yes | 16 | 5.3 |
| **Soil eating during current pregnancy** | | |
| A few times | 31 | 10.2 |
| Always | 62 | 20.4 |
| Most times | 63 | 20.7 |
| Never | 148 | 48.7 |

as lower risk, reducing early opportunities for iron supplementation. In addition, they could be feeding on urbanized dietary habits which low in iron-rich traditional foods.

Being single was associated with higher odds of having anemia as compared to being married. This is consistent with finding from studies conducted in Rwanda and Nigeria which also found that single pregnant women had higher odds of having anemia [31,32]. Married women might have social and financial support from their husbands which can promote

**Table 3. Factors independently associated with anaemia among pregnant refugees settled in the West Nile region of Uganda.**

| Variable | cOR | 95% CI | p-value | aOR | 95% CI | p-value |
|---|---|---|---|---|---|---|
| Age | | | | | | |
| 15–24 | 1 | | | 1 | | |
| 25–34 | 1.23 | (0.77-1.97) | 0.381 | 1.14 | (0.85-1.52) | 0.386 |
| ≥35 | 0.79 | (0.30-2.13) | 0.644 | 0.76 | (0.35-1.64) | 0.482 |
| Highest education level | | | | | | |
| Informal/Primary | 1 | | | 1 | | |
| Secondary/Tertiary | 1.08 | (0.96-1.22) | 0.203 | 1.13 | (1.10-1.16) | **<0.001** |
| Marital Status | | | | | | |
| Married | 1 | | | 1 | | |
| Separated/Widowed | 0.5 | (0.43-0.59) | **<0.001** | 0.51 | (0.46-0.58) | **<0.001** |
| Parity | | | | | | |
| 0 | 1 | (0.56-1.78) | 0.992 | 1.01 | (0.56-1.82) | 0.972 |
| 1 | 1 | | | 1 | | |
| 2–4 | 1.15 | (0.76-1.75) | 0.505 | 1.18 | (0.74-1.90) | 0.482 |
| >4 | 1.2 | (0.80-1.78) | 0.376 | 1.49 | (1.10-2.01) | **0.009** |
| Number of ANC visits | | | | | | |
| <4 | 1 | | | 1 | | |
| ≥4 | 1.27 | (0.90-1.80) | 0.173 | 1.34 | (0.93-1.93) | 0.118 |
| Positive mRDT | | | | | | |
| No | 1 | | | 1 | | |
| Yes | 1.17 | (0.06-24.25) | 0.921 | 1.23 | (0.05-33.13) | 0.903 |
| Soil eating during current pregnancy | | | | | | |
| A few times | 1 | | | 1 | | |
| Always | 0.88 | (0.47-1.64) | 0.682 | 0.88 | (0.42-1.84) | 0.741 |
| Most times | 0.75 | (0.25-2.24) | 0.601 | 0.73 | (0.22-2.39) | 0.605 |
| Never | 0.76 | (0.43-1.34) | 0.346 | 0.75 | (0.43-1.30) | 0.302 |
| Ulcers/Gastritis | | | | | | |
| No | 1 | | | 1 | | |
| Yes | 0.99 | (0.50-1.94) | 0.973 | 0.9 | (0.53-1.52) | 0.684 |
| HIV status | | | | | | |
| Negative | 1 | | | 1 | | |
| Positive | 1.82 | (0.35-9.36) | 0.473 | 0.68 | (0.03-15.24) | 0.810 |
| Don't want to disclose | – | – | – | – | – | – |
| HPV or cervical cancer | | | | | | |
| No | 1 | | | 1 | | |
| Yes | 3.69 | (0.17-82.03) | 0.409 | 4.68 | (0.03-807.04) | 0.557 |
| Deworming in the last six months | | | | | | |
| No | 1 | | | 1 | | |
| Yes | 0.77 | (0.33-1.77) | 0.535 | 0.73 | (0.29-1.86) | 0.514 |

cOR: Crude Odds Ratio; aOR: Adjusted Odds Ratio; CI: Confidence Interval

regular iron supplementation and good nutrition as compared to single women. [31]. However, a study conducted in Rwanda among women of childbearing age found that married women were more likely to have anemia than single women [33].

Pregnant women with parity greater than 4 had higher odds of having anemia as compared to those with parity of one. Consistent with our finding, studies done elsewhere have revealed that high parity is associated with higher odds of having anemia during pregnancy [25,27,34]. Women with high parity have a higher risk of anemia because the cumulative nutritional demands of previous pregnancies can deplete a woman's body of essential nutrients like iron [25,34,35].

One of the strengths of this study is that it is among the first to assess anaemia among pregnant refugee women in Uganda. We also used both RDT and blood smear for malaria diagnosis. However, this study had some limitations. First, we did not collect data on gestational age thus couldn't compare anemia prevalence across pregnancy trimesters. Second, we did not assess hematinic use, which may have confounded our results. Third, we did not assess gestation age as anaemia prevalence varies by gestation age. Fourth, some data on the predictors were based on self-reports, which carries a risk of social desirability bias. Additionally, the data could be affected by recall and interviewer biases. Finally, the results of this study may not be generalizable to refugee women in the community who do not attend antenatal clinics, as this was a hospital-based study.

## Conclusions

Anemia in pregnancy is prevalent among pregnant refugee women in the West Nile Region of Uganda. Our findings underscore the need for targeted interventions such as health education about the consequences of anemia, promotion of early ANC attendance, and provision of iron and folate supplementation during pregnancy to potentially reduce the burden of anemia in this vulnerable population.

Policymakers should prioritize the development and implementation of policies that ensure consistent and adequate provision of nutritional supplements and integrate anemia screening into routine ANC services. Health education programs tailored to refugee communities can improve awareness and adherence to ANC visits and nutritional guidelines, leveraging community leaders and culturally appropriate communication strategies.

Future research should explore the underlying causes of anemia in this population, including dietary habits, healthcare access, and socio-economic factors. Humanitarian organizations should incorporate anemia prevention and management into their health programs, training healthcare workers in refugee camps and ensuring the availability of necessary supplements and medications to improve maternal and neonatal health outcomes.

## Supporting information

**S1 File. Dataset.**
(XLS)

## Acknowledgement

We acknowledge administrative support from the study sites, office of the prime minister and medical teams international.

## Author contributions

**Conceptualization:** Felix Bongomin, Winnie Kibone, Sarah Lebu, Phillip Musoke, Chimdi Muoghalu, Stephen Ochaya, Musa Manga.

**Data curation:** Felix Bongomin, Winnie Kibone, Ritah Nantale, Rachel Beardsley, Musa Manga.

**Formal analysis:** Felix Bongomin, Winnie Kibone, Ritah Nantale, Phillip Musoke, Rachel Beardsley, Stephen Ochaya, Musa Manga.

**Funding acquisition:** Felix Bongomin, Sarah Lebu, Stephen Ochaya.

**Investigation:** Felix Bongomin, Winnie Kibone, Sarah Lebu, Byron Awekonimungu, Stephen Ochaya, Musa Manga.

**Methodology:** Felix Bongomin, Winnie Kibone, Ritah Nantale, Sarah Lebu, Byron Awekonimungu, Phillip Musoke, Chimdi Muoghalu, Stephen Ochaya, Musa Manga.

**Project administration:** Felix Bongomin, Stephen Ochaya.

**Resources:** Felix Bongomin, Sarah Lebu.

**Software:** Felix Bongomin.

**Supervision:** Felix Bongomin, Winnie Kibone, Byron Awekonimungu, Phillip Musoke, Musa Manga.

**Validation:** Felix Bongomin, Winnie Kibone, Ritah Nantale, Phillip Musoke.

**Visualization:** Felix Bongomin, Winnie Kibone, Ritah Nantale, Daniel S. Ebbs.

**Writing – original draft:** Felix Bongomin, Winnie Kibone, Ritah Nantale, Sarah Lebu, Byron Awekonimungu, Phillip Musoke, Daniel S. Ebbs, Rachel Beardsley, Chimdi Muoghalu, Stephen Ochaya, Musa Manga.

**Writing – review & editing:** Felix Bongomin, Winnie Kibone, Ritah Nantale, Sarah Lebu, Byron Awekonimungu, Phillip Musoke, Daniel S. Ebbs, Rachel Beardsley, Chimdi Muoghalu, Stephen Ochaya, Musa Manga.

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
