## [Decision Letter · Decision Letter 0]

15 May 2025

Dear Dr. Bongomin,

Thank you for submitting your manuscript to PLOS ONE. After careful consideration, we feel that it has merit but does not fully meet PLOS ONE’s publication criteria as it currently stands. Therefore, we invite you to submit a revised version of the manuscript that addresses the points raised during the review process.

**ACADEMIC EDITOR:**

Provide more background on the refugee population (origin, duration in camps), healthcare structure, and the number of facilities serving refugees in Adjumani District.Clarify the selection method for the three health facilities specify if it was random, purposive, or otherwise and how the sample size was distributed among them.

Methods 

Explain if haematinics are routinely provided at ANC and whether their use was assessed.Clarify the procedures for malaria and stool testing, including why these were done and ensure results are reported.Justify the collection of anthropometric data (e.g., weight) and blood samples state how much was collected, how often, and include SOPs on sample handling and transport if done across sites.Explain how questionnaire data were matched with lab results and whether any systems were used to prevent data errors.

Results, clarity and consistency 

Account for the reduced sample size and report the response rate.Resolve inconsistencies in income and breadwinner data and align variable categories across tables (e.g., education and marital status).Clarify variable definitions such as “time at camp,” “soil eating,” and how diagnoses (e.g., gastritis, HPV/cancer) and deworming history were established.

Discussions and limitations 

Compare findings with studies among similar populations (i.e., pregnant refugees).Explain unexpected findings (e.g., higher anaemia in educated women).Discuss limitations of RDT-based malaria diagnosis and the omission of key variables like haematinic use.Add a brief section on the strengths of the study.

We look forward to receiving your revised manuscript.

Kind regards,

Henry Komakech, MHSR, B.A

Academic Editor

PLOS ONE

Journal Requirements:

4. Please include captions for your Supporting Information files at the end of your manuscript, and update any in-text citations to match accordingly. Please see our Supporting Information guidelines for more information: http://journals.plos.org/plosone/s/supporting-information .

Additional Editor Comments:

The manuscript would benefit from additional contextual details about the study setting and clearer descriptions of methods, including sampling procedures, data collection processes, and justification for measurements conducted. Please also ensure consistency between the methods and results sections by including findings for all procedures mentioned and clarifying any discrepancies in tables and terminology used.

The authors should clarify discrepancies and definitions in the tables, particularly regarding income sources, education, marital status, and newly introduced variables like ‘soil eating.’ Additionally, explanations should be provided for how certain health conditions and variables were assessed, and missing methodological details should be included to ensure alignment between the methods and results sections.

The authors should provide appropriate comparisons with anaemia prevalence among pregnant refugee women in similar contexts and offer a clear, logical explanation for the unexpected association between education and anaemia in their findings. Additionally, they should explain the absence of gestational age data, expand on the study’s limitations, and highlight its key strengths.

Reviewers' comments:

Reviewer's Responses to Questions

**Comments to the Author**

1. Is the manuscript technically sound, and do the data support the conclusions?

Reviewer #1: Yes

Reviewer #2: Yes

2. Has the statistical analysis been performed appropriately and rigorously?

Reviewer #1: I Don't Know

Reviewer #2: Yes

3. Have the authors made all data underlying the findings in their manuscript fully available?

Reviewer #1: No

Reviewer #2: Yes

4. Is the manuscript presented in an intelligible fashion and written in standard English?

Reviewer #1: Yes

Reviewer #2: Yes

Reviewer #1: General comments

The study addresses a critical public health issue in a vulnerable population and provides valuable localized data. There are however some issues the needs to be addressed or clarified to improve the quality of the manuscript.

Specific comments and recommendations

Study settings: The manuscript will benefit from more information on the setting eg – the geopolitical structure of the country, how healthcare delivery is structured in the country, who are the refugees and where did they migrate from, how long have they settled there, socio-economic context, etc.

What about the provision of routine haematinics, is it not part of the services offered at the ANC?

Line 116 – Please explain this process of ‘random’ selection of 3 facilities. How many facilities in all were available? Did you do simple, systematic, or cluster selection? Or were the 3 facilities rather purposively selected?

Line 117 – there is no reflection of Ayiri HC III in Table 1

Line 135: In order to avoid confusion with qualitative interviews, I would suggest authors rephrase the use of the term “face-to-face interview” to ‘survey’ or a more unambiguous term.

Anthropometry: Measurement of weight is mentioned. Why was it done? There was no reflection of this in the results.

Malaria test: Blood film microscopic examinations were done along with parasite density. There was no reflection of this in the results

Line 185 – There was no mention of stool analysis in the methods

Results

Line 191 – Please state the reason(s) why the final sample was less than the calculated sample size i.e. why were the others excluded from the final analysis

Table 1:

Please rephrase ‘Mother’s age’ to simple ‘Age’

Source of income – Please clarify and reconcile the fact that 261 had no source of income, yet everyone had ‘estimated monthly income’. Furthermore, 267 were the bread-winners despite the majority not having a source of income?

“Time at the camp’ – Does this mean that the refugees got to the area at different times? This underscores the earlier comment regarding more background information about the setting

Table 2:

Gastritis/Ulcer – how were these determined? Please include in the methods

“HPV infection or cancer” – how were these diagnoses determined?

“Dewormed in last 6 months” – Please clarify how this was determined, and include the definition in the relevant section of the methods

Table 3:

Highest education level: please clarify those described as ‘none’, are they the same as ‘informal’ on Table 1?

Marital status: Pls clarify the components of the ‘single’ marital group, did this include those who were described as ‘separated’ on Table 1

‘Soil eating…’ is appearing for the first time here in a multivariable analysis?

Discussion

Lines 251-254: You should rather proffer possible explanations for your own counter-intuitive finding instead of focusing on explaining the findings by others.

Limitations: The use of haematinics (yes or no) is a critical confounding variable not mentioned in the manuscript. Was this left out of the data collected? It appears this potentially compromised the validity of the findings.

The authors should also discuss the limitations associated with dependence on RDT for malaria diagnosis as presented in the results.

Reviewer #2: This study presents an investigation into the prevalence and severity of anaemia among pregnant refugee women settled in the West Nile region of Uganda, addressing a significant public health concern in a vulnerable population. The manuscript is clearly written, methodologically sound, and the findings are well presented. Nonetheless, several important areas require clarification and refinement. I have provided some constructive comments to improve the manuscript. Thank you

Abstract and Background

1. They are precise and well-written.

Methods

2. How did you randomly select the three facilities included in the study?

3. For the reader's context, could you provide more details about which areas of the Adjumani district have refugees and how many facilities, among which you chose, serve them?

4. Explain how the sample size was distributed among the three facilities.

5. Line 138: Provide references to the studies alluded to.

6. I note that complete blood count tests were all conducted at one facility (Mungula HCIV laboratory). Did you transport samples from other sites to this laboratory? If so, please provide a summary of the SOPs you followed (i.e vacutainers in which blood was collected, sample labelling, storage and transportation of the sample to the laboratory, time from collection to running the test etc). Please also attach the detailed SOPs used as a supplement file.

7. You collected 4 mLs of CBC and another 4 mLs for malaria testing, a total of 8 mLs? According to your write-up, it seems you bled the patient twice. Is it so? Include a blood sampling section in the methods and clearly explain the procedures related to this.

8. How did you synchronize patient records from the questionnaires to the test results from the laboratory? Did you use any system, was it pre-programmed to prevent mix-up of data. Please provide these details under your data collection procedures.

9. You mention that patients were provided results of stool analysis, but that has not been elaborated in the study procedures. Please provide it, and what was the stool analysis performed for?

Results

10. Line 191: Note the response rate of your study

11. In the study procedures, you performed microscopy and quantification of malaria parasites for participants, but none of this was reported in the results.

Discussion

12. Line 233-237: From the references, I note that the studies from Tanzania, Ethiopia, India, Sudan and Ghana are not explicitly among refugee pregnant women, unless otherwise. You need to do some comparisons with the prevalence of pregnant refugee women elsewhere.

13. Line 246-254: I found no explanation for why educated women in this study were likely to have anaemia, contrary to many other studies, as you noted. You instead gave a reason why educated women may have a lower prevalence of anaemia, as reported in different studies. Please provide a logical explanation for your finding.

14. Why were gestational ages not collected despite having access to the patients and their ANC card records as well?

15. Limitations of this study have not been exhaustively stated.

16. Please also state the strengths of your study.

---

## [Author Response · Author response to Decision Letter 1]

29 Jun 2025

PONE-D-24-37952

Anemia prevalence and severity among pregnant refugee women settled in the west Nile region, Uganda

PLOS ONE

Dear Dr. Bongomin,

Thank you for submitting your manuscript to PLOS ONE. After careful consideration, we feel that it has merit but does not fully meet PLOS ONE’s publication criteria as it currently stands. Therefore, we invite you to submit a revised version of the manuscript that addresses the points raised during the review process.

ACADEMIC EDITOR:

Study context and setting

• Provide more background on the refugee population (origin, duration in camps), healthcare structure, and the number of facilities serving refugees in Adjumani District.

Authors response: This has been updated

• Clarify the selection method for the three health facilities specify if it was random, purposive, or otherwise and how the sample size was distributed among them.

Authors response: This was random as indicated in the methods section and samples distributed proportionately.

Methods

• Explain if haematinics are routinely provided at ANC and whether their use was assessed.

Authors response: Yes, and this has been indicated. However, we didn’t assess this as was assumed to be universal. This is a limitation we have now acknowledged.

• Clarify the procedures for malaria and stool testing, including why these were done and ensure results are reported.

Authors response: Procedure for malaria is explained in the methods. Details of stool examinations and STHIs is a subject of a subsequent sub study being written up.

• Justify the collection of anthropometric data (e.g., weight) and blood samples state how much was collected, how often, and include SOPs on sample handling and transport if done across sites.

Authors response: Thank you for this observation, we have updated Table 1 to include the mean and standard deviation of participants’ weight.

• Explain how questionnaire data were matched with lab results and whether any systems were used to prevent data errors.

Authors response: Thank you for this comment, participants were assigned a unique ID, which was used consistently across questionnaires and lab forms. We have updated this under study measurements in the methods section, line number 171

Results, clarity and consistency

• Account for the reduced sample size and report the response rate.

Authors response: This has been addressed.

• Resolve inconsistencies in income and breadwinner data and align variable categories across tables (e.g., education and marital status).

Authors response: This has been addressed.

• Clarify variable definitions such as “time at camp,” “soil eating,” and how diagnoses (e.g., gastritis, HPV/cancer) and deworming history were established.

Authors response: This has been addressed.

Discussions and limitations

• Compare findings with studies among similar populations (i.e., pregnant refugees).

Authors response: Thank you for this comment, we have included comparative data from studies among refugee pregnant women in Ethiopia, Rwanda and Somalia. Line number 242-243

• Explain unexpected findings (e.g., higher anaemia in educated women).

Thank you. This has been revised. We now suggest that educated women may face occupational stress, delayed ANC initiation, and urbanized dietary habits low in iron-rich traditional foods. This could explain the unexpected association. Line number 261-264

• Discuss limitations of RDT-based malaria diagnosis and the omission of key variables like haematinic use.

We have included additional limitations, lack of data on haematinic use, limited parasite detection using RDT, and potential recall bias. Line number 278-287

• Add a brief section on the strengths of the study.

Authors response: Thank you, this has been done. Line number 278

We look forward to receiving your revised manuscript.

Kind regards,

Henry Komakech, MHSR, B.A

Academic Editor

PLOS ONE

Journal Requirements:

Additional Editor Comments:

The manuscript would benefit from additional contextual details about the study setting and clearer descriptions of methods, including sampling procedures, data collection processes, and justification for measurements conducted. Please also ensure consistency between the methods and results sections by including findings for all procedures mentioned and clarifying any discrepancies in tables and terminology used.

The authors should clarify discrepancies and definitions in the tables, particularly regarding income sources, education, marital status, and newly introduced variables like ‘soil eating.’ Additionally, explanations should be provided for how certain health conditions and variables were assessed, and missing methodological details should be included to ensure alignment between the methods and results sections.

The authors should provide appropriate comparisons with anaemia prevalence among pregnant refugee women in similar contexts and offer a clear, logical explanation for the unexpected association between education and anaemia in their findings. Additionally, they should explain the absence of gestational age data, expand on the study’s limitations, and highlight its key strengths.

Reviewers' comments:

Reviewer's Responses to Questions

Comments to the Author

1. Is the manuscript technically sound, and do the data support the conclusions?

Reviewer #1: Yes

Reviewer #2: Yes

2. Has the statistical analysis been performed appropriately and rigorously?

Reviewer #1: I Don't Know

Reviewer #2: Yes

3. Have the authors made all data underlying the findings in their manuscript fully available?

Reviewer #1: No

Reviewer #2: Yes

4. Is the manuscript presented in an intelligible fashion and written in standard English?

Reviewer #1: Yes

Reviewer #2: Yes

5. Review Comments to the Author

We thank the reviewers for the insightful comments and valuable suggestions that have improved the clarity and quality of our manuscript. Below are our detailed responses.

Reviewer #1: General comments

The study addresses a critical public health issue in a vulnerable population and provides valuable localized data. There are however some issues the needs to be addressed or clarified to improve the quality of the manuscript.

Specific comments and recommendations

Study settings: The manuscript will benefit from more information on the setting eg – the geopolitical structure of the country, how healthcare delivery is structured in the country, who are the refugees and where did they migrate from, how long have they settled there, socio-economic context, etc.

What about the provision of routine haematinics, is it not part of the services offered at the ANC?

Authors response: This has been addressed.

Line 116 – Please explain this process of ‘random’ selection of 3 facilities. How many facilities in all were available? Did you do simple, systematic, or cluster selection? Or were the 3 facilities rather purposively selected?

The facilities were purposively selected based on levels and access to labs.

Line 117 – there is no reflection of Ayiri HC III in Table 1

Authors response: This has been revised. It is the 1st row.

Line 135: In order to avoid confusion with qualitative interviews, I would suggest authors rephrase the use of the term “face-to-face interview” to ‘survey’ or a more unambiguous term.

Thank you for the suggestion, this has been done. Line number 135; Data were collected through surveys administered by trained research assistants on information on maternal characteristics such as age, gravidity, education level, occupation, marital status, HIV status, and the number of ANC visits in the current pregnancy.

Anthropometry: Measurement of weight is mentioned. Why was it done? There was no reflection of this in the results.

Thank you for this observation, we have updated Table 1 to include the mean and standard deviation of participants’ weight.

Malaria test: Blood film microscopic examinations were done along with parasite density. There was no reflection of this in the results

Authors response: This has been addressed. We have included Blood smear results

Line 185 – There was no mention of stool analysis in the methods

Authors response: we have removed this.

Results

Line 191 – Please state the reason(s) why the final sample was less than the calculated sample size i.e. why were the others excluded from the final analysis

Thank you for your comment, we were able to enroll 304 participants due to logistic issues within the settelemnt during the study. This has been updated in the methods section.

Table 1:

Please rephrase ‘Mother’s age’ to simple ‘Age’

Thank you, this has been revised accordingly. Line number: 203, Table 1

Source of income – Please clarify and reconcile the fact that 261 had no source of income, yet everyone had ‘estimated monthly income’. Furthermore, 267 were the bread-winners despite the majority not having a source of income?

Authors response: We agree this was a difficult question since most relied on reliefs from OPM and partners. However, we probed as much as possible and we have reported what the participants were able to tell us.

“Time at the camp’ – Does this mean that the refugees got to the area at different times? This underscores the earlier comment regarding more background information about the setting

Authors response: Yes, they came at different times

Table 2:

Gastritis/Ulcer – how were these determined? Please include in the methods

“HPV infection or cancer” – how were these diagnoses determined?

“Dewormed in last 6 months” – Please clarify how this was determined, and include the definition in the relevant section of the methods

Authors response: This were self-report and refelected now in the medthods.

Table 3:

Highest education level: please clarify those described as ‘none’, are they the same as ‘informal’ on Table 1?

Thank you, this has been revised.

Marital status: Pls clarify the components of the ‘single’ marital group, did this include those who were described as ‘separated’ on Table 1

‘Soil eating…’ is appearing for the first time here in a multivariable analysis?

Under Table 3, the “single” category includes those widowed, or separated. We have clarified this in the Table 3. Soil eating has now been included in the descriptive analysis in Table 2.

Discussion

Lines 251-254: You should rather proffer possible explanations for your own counter-intuitive finding instead of focusing on explaining the findings by others.

Thank you. This has been revised. We now suggest that educated women may face occupational stress, delayed ANC initiation, and urbanized dietary habits low in iron-rich traditional foods. This could explain the unexpected association. Line number 261-264

Limitations: The use of haematinics (yes or no) is a critical confounding variable not mentioned in the manuscript. Was this left out of the data collected? It appears this potentially compromised the validity of the findings.

The authors should also discuss the limitations associated

---

## [Editor Report · Decision Letter 1]

24 Jul 2025

Anemia prevalence and severity among pregnant refugee women settled in the west Nile region, Uganda

PONE-D-24-37952R1

Dear Dr. Bongomin,

We’re pleased to inform you that your manuscript has been judged scientifically suitable for publication and will be formally accepted for publication once it meets all outstanding technical requirements.

Kind regards,

Henry Komakech, MHSR, B.A

Academic Editor

PLOS ONE

---

## [Editor Report · Acceptance letter]

PONE-D-24-37952R1

PLOS ONE

Dear Dr. Bongomin,

I'm pleased to inform you that your manuscript has been deemed suitable for publication in PLOS ONE. Congratulations! Your manuscript is now being handed over to our production team.

Kind regards,

on behalf of

Dr. Henry Komakech

Academic Editor

PLOS ONE